# Scaling Laws for Adversarial Attacks on Language Model Activations and Tokens

**Stanislav Fort**
*Independent Researcher*
Prague, Czech Republic

## Abstract

We explore a class of adversarial attacks targeting the activations of language models to derive upper-bound scaling laws on their attack susceptibility. By manipulating a relatively small subset of model activations, $a$, we demonstrate the ability to control the exact prediction of a significant number (in some cases up to 1000) of subsequent tokens $t$. We empirically verify a scaling law where the maximum number of target tokens predicted, $t_{\max}$, depends linearly on the number of tokens $a$ whose activations the attacker controls as $t_{\max} = \kappa a$. We find that the number of bits the attacker controls on the input to exert a single bit of control on the output (a property we call *attack resistance $\chi$*) is remarkably stable between $\approx 16$ and $\approx 25$ over orders of magnitude of model sizes and between model families. Compared to attacks directly on input tokens, attacks on activations are predictably much stronger, however, we identify a surprising regularity where one bit of input steered either via activations or via tokens is able to exert a surprisingly similar amount of control over the model predictions. This gives support for the hypothesis that adversarial attacks are a consequence of dimensionality mismatch between the input and output spaces. A practical implication of the ease of attacking language model activations instead of tokens is for multi-modal and selected retrieval models. By using language models as a controllable test-bed to study adversarial attacks, we explored input-output dimension regimes that are inaccessible in computer vision and greatly extended the empirical support for the dimensionality theory of adversarial attacks.

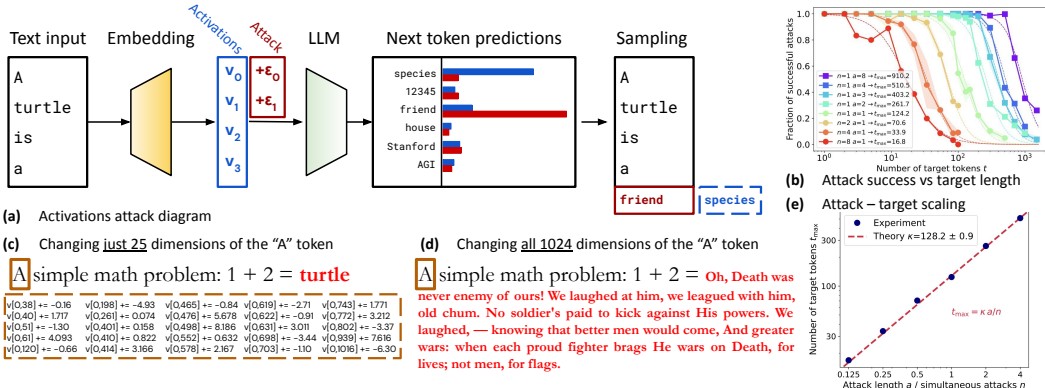

Figure 1: (a) A diagram showing an attack on the activations (blue vectors) of a language model that leads to the change of the predicted next token from *species* to *friend*. The maximum number of tokens whose values can be dictated precisely, $t_{\max}$, can be read off the attack success rate plots (b), and it scales linearly with the number of attack tokens $a$ (e). An example of attacking just 25 dimensions (out of 1024) of the first token embedding is shown in (c), leading to the prediction of *turtle*, while attacking all 1024 dimension is shown in (d), leading to reciting a 40-token poem.

# 1 INTRODUCTION

Adversarial attacks pose a major challenge for deep neural networks, including state-of-the-art vision and language models. Small, targeted perturbations to the model input can have very large effects on the model outputs and behaviors. This raises concerns around model security, safety and reliability, which are increasingly practically relevant as machine learning systems get deployed in high-stakes domains such as medicine, self-driving, and complex decision making. While most work has focused on attacking image classifiers, where adversarial examples were first identified (Szegedy et al., 2013), large language models (LLMs) both 1) provide a natural, controllable test-bed for studying adversarial attacks more systematically and in otherwise inaccessible regimes, and 2) are of a great importance on their own, since they are increasingly becoming a backbone of many advanced AI applications.

An adversarial attack on an image classifier is a small, targeted perturbation added to its continuous input (e.g. an image) that results in a dramatic change of the resulting classification decision from one class to another, chosen by the attacker. Working with language models, we immediately face two core differences: their input is a series of discrete tokens, not a continuous signal, and the model is often used in an autoregressive way (popularly referred to as "generative") to generate a continuation of a text, rather than classification. In this paper, we side step the discrete input issue by working with the continuous model activations (sometimes referred to as the residual stream (Elhage et al., 2021)) that the discrete tokens get translated to by the embedding layer at the very beginning of the model. We resolve the second issue by viewing a language model as a classifier from the continuous activations (coming from input tokens) to a discrete set of $t$-token continuations that are drawn from $V^t$ possibilities ($V$ being the vocabulary size of the model). We compare these *activation attacks* to *token substitution attacks* as well.

We hypothesize that the mismatch between the dimensions of the input space (that the attacker can control) and the output space is a key reason for adversarial susceptibility of image classifiers, concretely the much larger input dimension over the output one. A similar argument can be traced through literature (Goodfellow et al., 2015; Abadi et al., 2016; Ilyas et al., 2019; Fort & Lakshminarayanan, 2024).

Does a significant vulnerability to activation attacks pose a practical vulnerability given that a typical level of access to a large language model stays at the level of input tokens (especially for commercial models)? While token access is standard, there are at least two very prominent cases where an attacker might in fact access and control the activations directly: 1) Retrieval, and 2) Multi-modal models. **Retrieval**: Borgeaud et al. (2022) uses a database of chunks of documents that are on the fly retrieved and used by a language model. Instead of injecting the retrieved pieces of text directly as *tokens*, a common strategy is to encode and concatenate them with the prompt activations directly, skipping the token stage altogether. This gives a direct access to the activations to whoever controls the retrieval pipeline. **Multi-modal models**: Alayrac et al. (2022) insert embedded images as activations in between text token activations to allow for a text-image multi-modal model. Similarly to the retrieval case, this allows an attacker to modify the activations directly. It is likely that similar approaches are used by other vision-LMs as well as LMs enhanced with other non-text modalities, posing a major threat.

**Related work.** Understanding a full LLM-based system instead of just analyzing the main model has been highlighted in Debenedetti et al. (2023) as very relevant to security, as the add-ons on top of the main LLM open additional attack surfaces. Similar issues have been highlighted among open questions and problems in reinforcement learning from human feedback (RLHF, Bai et al. (2022)) (Casper et al., 2023). Modifying model activations directly was also done in Zou et al. (2023a).

Scaling laws for language model performance, as a function of the parameter count and the amount of data, have been identified in Kaplan et al. (2020), refined in Hoffmann et al. (2022) and worked out for sparsely-connected models in Frantar et al. (2023). Similar empirical dependencies are also frequent in machine learning beyond performance prediction, e.g. the dependence between classification accuracy and near out-of-distribution robustness (Fort et al., 2021). Scaling laws have been identified in biological neural networks, for example between the number of neurons and the mass of the brain in mammals (Herculano-Houzel, 2012), and birds (Kabadayi et al., 2016), showing that performance scales with the $\log$ of the number of pallial or cortical neurons.

Using activation additions (Turner et al., 2023) shows some level of control over model outputs. A broad exploration and literature on model jail-breaking can also be seen in the light of adversarial attacks. Zou et al. (2023b) uses a mixture of greedy and gradient-based methods to find token suffixes that "jail-break" LLMs. Wang et al. (2023) claims that larger models are easier to jailbreak as a consequence of being better at following instructions. Attacks on large vision models, such as CLIP (Radford et al., 2021) are discussed in e.g. Fort (2021b;a), and the first successful transferable vision attacks on vLLMs are demonstrated in (Fort & Lakshminarayanan, 2024).

## 2 THEORY

**Problem setup.** Given an input string that gets tokenized into a series of $s$ integer-valued tokens $S$ (each drawn from a vocabulary of size $V$ as $S_i \in \{0, 1, \ldots V - 1\}$), a language model $f$ can be viewed as a classifier predicting the probabilities of the next-token continuation of that sequence over the vocabulary $V$. Were we to append the predicted token to the input sequence, we would be running the language model in its typical, autoregressive manner. Given this new input sequence, we could get the next token after, and repeat the process for as long as we need to.

Let's consider predicting a $t$-token sequence that would follow the input context. There are $V^t$ such possible outputs. This process is now mapping an $s$-token input sequence, for which there are $V^s$ many combinations, into its $t$-token continuation, for which there are $V^t$ combinations. Out of the $s$ input tokens, we could choose a subset of $a \leq t$ that would be the *attack* tokens the attacker can control. In this setup, we have a controllable classification experiment where the dimension of the input space $a$ (that the attacker controls), and the dimension of the target space, $t$, that they wish to determine the outputs in, are experimental dials that we can set and control explicitly.

**Attacking activation vectors.** To match the situation to the usual classification setup, we need a continuous input space. Instead of studying the behavior of the full language model mapping $s$ (or $a$) discrete tokens into probabilities of $t$-token sequences, we can first turn the input sequence $S$ into activation vectors, each of dimension $d$, (sometimes referred to as the *residual stream* (Henighan et al., 2023)) as $f_{\text{before}}(S \in \{0, 1, \ldots, V - 1\}^s) = v \in \mathbb{R}^{s \times d}$, and then propagate these activations through the rest of the network as $f_{\text{after}}(v)$.

The goal of the attacker is to come up with a perturbation $P$ to the first $a$ token activations (an arbitrary choice) within the vector $v$ such that the argmax autoregressive sampling from the model would yield the target sequence $T$ as the continuation of the input sequence $S$. Practically, this means that we can imagine computing the activations $v$ from the input sequence $S$ by the embedding layer, adding the perturbation $P$ to it, and passing it on through the rest of the model to get the next token logits. If the attack is successful, then $\text{argmax} f_{\text{after}}(v + P) = T_0$. For the j[th] target token, we get the activations of the input string $S$ concatenated with $(j - 1)$ tokens of the target sequence, $f_{\text{before}}(S + T[: j])$, add the perturbation $P$ (that does not affect the activations of more than the first $a$ tokens $a \leq |S| \leq |S + T[: j]|$), and for a successful attack obtain the prediction of the next target token as $\text{argmax} f_{\text{after}}(f_{\text{before}}(S + T[: j]) + P) = T_j$. What is described here is a success condition for the attack $P$ towards the target sequence $T$ rather than a process to actually compute it practically, which is detailed in Section 3.

**Input and output space dimensions.** In a typical image classification setting, the number of classes is low, and consequently so is the dimension of the output space compared to the input space. For example, CIFAR-10 and CIFAR-100 have 10 and 100 classes respectively (Krizhevsky et al., a;b) (with $32 \times 32 \times 3 = 3072$-dimensional images), ImageNet has 1,000 classes (Deng et al., 2009), and ImageNet-21k 21,000 ($\log_2(21,000) \approx 14$) (Ridnik et al., 2021) with $224 \times 224 \times 3 = 150,528$-dimensional images.

In comparison, predicting a single token output for a language model already gives us $V \approx 50,000$ classes (for the tokenizer used in our models, typical numbers are between 10,000, and several 100,000s), and moving to $t$-token continuations gives us an exponential control over the number of classes. For this reason, using a language model as a controllable test-bed for studying adversarial examples is very useful. Firstly, it allows us to control the output space dimension, and secondly, it opens up output spaces of much higher dimensions than would be accessible in standard computer vision problems. In our experiments, we study target sequences of up to $t \approx 1000$ tokens, giving us $V^t \approx 2^{16000}$ options or *effective* classes in our "classification" problem. For our LLM experiments,

we actually get to realistic regimes in which the dimension of the space of inputs the attacker controls is much lower than the dimension of the space of outputs.

The attacker controls a part of the activation vector $v$ with a perturbation $P$ that has non-zero elements in the first $a$ token activations. $a$ determines the expressivity of the attack and therefore the attack strength. Unlike an attack on the discrete input tokens, each drawn from $V$ possibilities, controlling a $d$-dimensional vector of floating point numbers per token, each number of 16 bits itself, offers a vastly larger dimensionality to the attacker (although of course the model might not be utilizing the full 16 bits after training, signs of which we see in Section 4). There are $2^{16^d}$ possible single token activation values, compared to just $V$ for tokens. For example, for $d = 512$ and $V = 50000$ (typical numbers), $2^{16^d} = 2^{2048}$, while $V \approx 2^{16}$.

**Scaling laws.** The core hypothesis is that the ability to carry a successful adversarial attack depends on the ratio between the dimensions of the input and output spaces. The success of attacking a language model by controlling the activation vectors of the first $a$ tokens, hoping to force it to predict a specific $t$-token continuation after the context, should therefore involve a linear dependence between the two. Let us imagine a $t_{\max}$, the maximum length of a target sequence the attacker can make the model predict with $a$ attack tokens of activations. The hypothesized dependence is $t_{\max} \propto a$, or

$$t_{\max} = \kappa a, \tag{1}$$

where the specific scaling constant $\kappa$ is the *attack multiplier*, and tells us how many tokens on the output can a single token worth of activations on the input control. We can test the scaling law in Eq. 1 by observing if the maximum attack length $t$ scales linearly with the attack length $a$. The specific attack strength $\kappa$ is empirically measured and specific to each model.

If we were to use only a fraction $f$ of the activation vector dimensions in the $a$ attack tokens, the effective dimension the attacker controls would equivalently decrease by a factor of $f$. Therefore the revised scaling law would be $t_{\max} = \kappa f a$. We verify that the effect of the fraction of dimensions on the successful target length is the same as varying the attack length $a$. $t_{\max} = \kappa f a$.

In Fort (2023), the authors develop a single perturbation $P$ called a *multi-attack* that is able to change the classification of $n$ different images to $n$ arbitrary classes chosen by the attacker. This effectively increases the dimension of the output space by a factor of $n$, while keeping the attack dimension of $P$ constant, which is a useful fact for us. We run experiments in this setup as well, where we want a single activation perturbation $P$ to continue a context $S_1$ by a target sequence $T_1$, $S_2$ by $T_2$, and so on all the way to $S_n$ to $T_n$. The dimension of the output space increases by a factor $n$, and therefore the maximum target length $t_{\max}$ now scales as $t_{\max} n = \kappa f a$. The revised scaling law is therefore $t_{\max} = \kappa f a / n$.

The attack strength $\kappa$ is model specific and empirically determined. However, our geometric theory suggests that it should be linearly dependent on the dimension of the activations of the model. Let us consider a simple model where $\chi$ bits of control on the input are needed to determine a single bit on the output, and let us call $\chi$ the *attack resistance*. For a vocabulary of size $V$, each output token is specified by $\log_2 V$ bits. A single token has an activation vector specified by $d$ $p$-bit precision floating point numbers. There are therefore $dp$ bits the attacker controls by getting a hold of a single token of activations. The attack strength $\kappa$, which is the number of target tokens the attacker can control with a single token of an attack activation, should therefore be $\chi \kappa \log_2 V = dp$. We assume $\chi$ to be constant between models (although adversarial training probably changes it), and therefore our theory predicts that $\kappa = \frac{dp}{\chi \log_2 V}$ for a fixed attack resistance $\chi$. For a fixed numerical precision and vocabulary size, the resulting scaling is $\kappa \propto d$, i.e. the attack strength is directly proportional to the dimension of the activation vector (also called the residual stream), and we observe this empirically in e.g. Table 1. Having obtained empirically measured values of $\kappa$, we can estimate the attack resistance $\chi$, which we do in Section 4. In the simplest setting, we would expect $\chi = 1$, which would mean that a single dimension of the input controls a single dimension of the output. The specifics of the way input and output spaces map to each other are likely complex and the reason for why $\chi > 1$.

**Comparison to token-level substitution attacks.** As a comparison, we tried looking for token substitution attacks as well, which are significantly less expressive. For those, the attacker can change the first $a$ integer-valued tokens on the input (instead of their high-dimensional activations; the geometric of the input manifolds is discussed in e.g. (Fort et al., 2022)), trying to make the model

produce the attacker-specified $t$-token continuation as before. If the geometric theory from Section 2 holds, the prediction would be that the attack strength would go down by a factor proportional to the reduction in the dimension of the input space the attacker controls. Going from $adp$ bits of control to $a \log_2 V$ bits corresponds to a $dp/\log_2 V$ reduction, which means that the attack multiplier $\kappa_{\text{token}}$ for the token substitution method is predicted to be related to the activation attack strength $\kappa$ as $\kappa_{\text{token}} = \kappa \frac{\log_2 V}{dp}$. For $d = 512$ and $V = 50000$, the $\frac{\log_2 V}{dp} \approx 2 \times 10^{-3}$. Our results are shown in Section 4, specifically in Table 2. In addition, it is very likely that the full 16 bits of the activation dimensions are not used, and that instead we should be using a $p_{\text{effective}} < p$.

## 3 METHOD

**Problem setup.** A language model takes a sequence of $s$ integer-valued tokens $S = [S_1, S_2, \ldots, S_s]$, each drawn from a vocabulary of size $V$, $S_i \in \{0, 1, \ldots, V-1\}$, and outputs the logits $z$ over the vocabulary for the next-word prediction $z \in \mathbb{R}^V$. These are the unscaled probabilities that turn into probabilities as $p = \text{softmax}(z)$ over the vocabulary dimension. As described in Section 2, we primarily work with attacking the continuous model activations rather than the integer-valued tokens themselves.

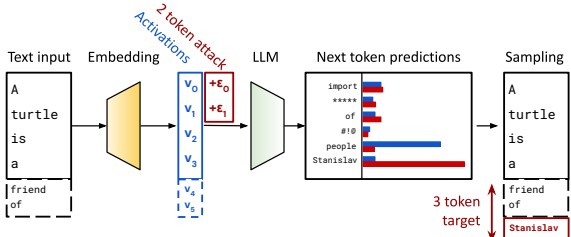

Figure 2: A diagram showing the $t = 3$ multi-token target prediction after an attack on $a = 2$ token activations.

The tokens $S$ are first passed through the embedding layer of the language model, producing a vector $v$ of $d$ dimensions per token, $v = f_{\text{before}}(S)$. (These are the activations that the attacker can modify by adding a perturbation vector $P$.) After that, the activations pass through the rest of the language model, as $f_{\text{after}}(v) = z$, to obtain the logits for the next-token prediction. The full language model mapping tokens to logits would be $f_{\text{after}}(f_{\text{before}}(S)) = z$, we just decided to split the full function $f : S \to z$ into the two parts, exposing the activations for an explicit manipulation. The split need not happen after the embedding layer, but rather after $L$ transformer layers of the model itself. While we ran exploratory experiments with splitting later in the model, we nonetheless performed all our detailed experiments with activations directly after the embedding layer.

**An attack on activations.** The attacker controls the activation perturbation $P$ that gets added to the vector $v$ in order to modify the next-token logits towards the token desired by the attacker. The modified logits are $z' = f_{\text{after}}(f_{\text{before}}(S) + P)$, where P, the attack vector, is of the shape $P \in \mathbb{R}^{s \times d}$ (the same as $v$), however, we allow the attacker to control only the activations of the first $a \leq s$ tokens. This is an arbitrary choice and other variants could be experimented with. The attack comprises the first $a$ tokens, leaving the remaining $s - a$ tokens separating the attack from its target. We experiment with the effect of this separation in Section G.

While the attacker controls the activations of the first $a$ tokens in the context, the model as described so far deals with affecting the prediction of the next token immediately after the $s$ tokens in the context. As discussed in Section 2, we want to predict $t$-token continuations instead.

**Loss evaluation and optimization.** To evaluate the loss $\mathcal{L}(S, T, P)$ of the attack $P$ on the context $S$ towards the target multi-token prediction $T$, we compute the standard language modeling cross-entropy loss, with the slight modification of adding the perturbation vector $P$ to the activations after the embedding layer. The algorithm is shown in Figure 8.

To find an adversarial attack, we first choose a fixed, randomly sampled context of size $S$, define the attack length (in tokens) $a$, choose a target length $t$ (in tokens), and a random string of $t$ tokens as the target sequence $T$. We then use the Adam optimizer (Kingma & Ba, 2014) and the gradient of the loss specified in Figure 8 with respect to the attack vector, $P$, as $g = \frac{\partial \mathcal{L}(S,T,P)}{\partial P}$. Using the gradient directly is the same technique as used in the original Szegedy et al. (2013), however, small modifications, such as keeping just the gradient signs (Goodfellow et al., 2015) are readily available as well. Decreasing the language modeling loss $\mathcal{L}$ by changing the activation attack $P$ translates into

making the model more likely to predict the desired $t$-token continuation $T$ after the context $S$ by changing the activations of the first $a$ tokens. We stop the experiment either 1) after a predetermined number of optimization steps, or 2) once the $t$-token target continuation $T$ is the argmax sampled continuation of the context $S$, by which we define a successful attack.

**Estimating the attack multiplier $\kappa$.** Our goal is to empirically measure under what conditions adversarial examples are *generally* possible and easy to find. We use random tokens sampled uniformly both for the context $S$ as well as the targets $T$ to ensure fairness. For a fixed attack length $a$ and a context size $s \geq a$, we sweep over target lengths $t$ in a range from 1 to typically over 1000 in logarithmic increments. For each fixed $(a, t)$, we repeat an experiment where we generate random context tokens $S$, and random target tokens $T$, and run the optimization at learning rate $10^{-1}$ for 300 steps. The success of each run is the fraction of the continuation tokens $T$ that are correctly predicted using argmax as the sampling method. This gives us, for a specific context length $s$, attack length $a$, and a target length $t$ an estimate of the attack success probability $p(a, t)$. The plot of this probability can be seen in e.g. Figure 3a and Table 1 refers to the appropriate figure for each model. The $p(a, t)$ are our main experimental result and we empirically estimate them for a range of language models and context sizes $s$.

For short target token sequences, the probability of a successful attack is high, and for long target sequences, the attacker is not able to control the model output sufficiently, resulting in a low probability. To $p(a, t)$ we fit a sigmoid curve of the form $\sigma(t, \alpha, \beta) = 1 - (1 + \exp(-\alpha(\log(t) - \log(t_{\max}))))$ and read-off the best fit value of $t_{\max}$, for which the success rate of the attack of length $a$ falls to 50%. In our scaling laws, we work with these values of $t_{\max}$, however, the 50% threshold is arbitrary and can be chosen differently.

Empirically, the read-off value of the 50% attack success threshold $t_{\max}$ depends linearly on the number of attack tokens $a$ whose activations the attacker can modify. The linearity of the relationship can be seen in e.g. Figure 6. As described in Section 2, we call the constant of proportionality the *attack multiplier $\kappa$*, and it relates the attack and target lengths as $t_{\max} = \kappa a$.

We empirically observe that the attack multiplier $\kappa$ depends linearly on the dimension of the model activations used even across different models. We also theoretically expect this in Section 2. The *attack resistance $\chi$*, defined in Equation 2, can be calculated from the estimated attack multiplier $\kappa$ with the knowledge of the model activation numerical precision $p$ (in our case 16 bits in all cases), activation dimension $d$ (varied from 512 to 2560), and vocabulary size $V$ (around 50,000 for all experiments) as $\chi = \frac{dp}{\kappa \log_2 V}$. We provide these estimates in Table 1.

In Fort (2023) adversarial *multi-attacks* are defined and described. They are attacks in which a single adversarial perturbation $P$ is able to convert $n$ inputs into $n$ attacker-chosen classes. We ran a similar experiment where a single adversarial perturbation $P$ can make the context $S_1$ complete as $T_1$, $S_2$ as $T_2$, and all the way to $S_n$ to $T_n$. This effectively decreases the attack length $a$ by a factor of $n$, or equivalently increases the target length by the same factor, as discussed on dimensional grounds in Section 2. Practically, we accumulate gradients over the $n$ $(S, T)$ pairs before taking an optimization step on $P$. We study multi-attacks from $n = 1$ (the standard attack) to $n = 8$.

Another modification described in Section 2 is to use only a random, fixed fraction $f$ of the dimensions of the activation vector $P$. Its effect is to change the effective attack length from $a$ to $fa$, and we choose the mask uniformly at random.

**Token substitution attacks.** To compare the attack on activations to the more standard attack on the input tokens themselves, we used a greedy, token-by-token, exhaustive search over all attack tokens $a$ at the beginning of the context $S$. For a randomly chosen context $S$, an attack length $a \leq |S|$, and a randomly chosen sequence of target tokens $T$, we followed the algorithm in Figure 12 to find the first $a$ tokens of the context that maximize the probability of the continuation $T$. By greedily searching over all $V$ possible tokens, each attack token at a time, we can guarantee convergence in $aV$ steps. We repeat this experiment over different values of the attack length $a$, and random contexts $S$ and targets $T$ of length $t = |T|$, obtaining a similar $p_{\text{token}}(a, t)$ curve as for the activation attacks. We fit the Eq. 3 sigmoid to it, extracting an equivalent attack multiplier $\kappa_{\text{token}}$, characterizing how many tokens on the output can a single token on the input influence (the result is, unlike for the activation attacks, much smaller than 1, of course).

**Attack and target separation within the context.** The further the attack is from the target, the less effective it might be. We therefore experiment with different sizes of the context $S$ that separate the first $a$ token activations of the attack from the target tokens after $S$. We estimate an attack multiplier for each $s$, getting a $\kappa(s)$ curve that we show in Figure 3c. The attack multiplier $\kappa$ looks constant up to a point (around 100 tokens of context) and then decreases linearly in $\log(s)$. We therefore fit a simple function of this form to our data in Figure 3c.

## 4 RESULTS AND DISCUSSION

We have been using the `EleutherAI/pythia` series of Large Language Models (Biderman et al., 2023) based on the GPT-NeoX library (Andonian et al., 2021; Black et al., 2022) from Hugging Face[1]. A second suite of models we used is the `microsoft/phi-1`[2] (Li et al., 2023). Finally, we used a single checkpoint of `roneneldan/TinyStories`[3] presented in Eldan & Li (2023). We ran our experiments on a single A100 GPU on a Google Colab.

For finding the adversarial attacks on activations, we used the Adam optimizer (Kingma & Ba, 2017) at a learning rate of $10^{-1}$ for 300 optimization steps, unless explicitly stated otherwise. Our activations were all in the `float16` format, and the model vocabulary sizes were all very close to $V \approx 50,000$. For the input context as well as our (multi-)token target sequences, we sampled random tokens from the vocabulary uniformly. When using only a subset of the activations, as described in Section 3, we choose the dimensions at random uniformly.

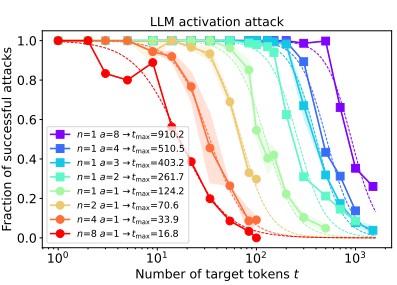

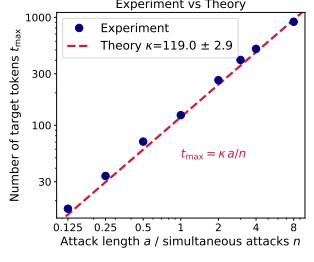

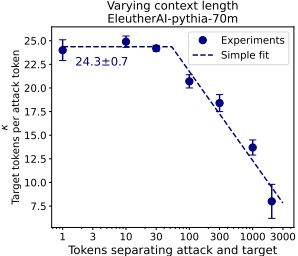

(a) Probability of attack success as function of the number of target tokens $t$ for different lengths of the attack $a$.

(b) The maximum number of successfully converted target tokens $t_{\max}$ as a function of the the attack length $a$.

(c) The effect of tokens separating the attack and the target. Up to 100 tokens of separation, the attack multiplier (strength) does not diminish.

Figure 3: A summary of adversarial attacks on activations of EleutherAI/pythia-1.4b-v0. Only experiments varying the attack length $a$ (in tokens whose activations the attacker controls) and the multiplicity of context and target pairs the attack has to succeed on, $n$, are shown. The estimated attack multiplier is $\kappa = 119.0 \pm 2.9$ which means that controlling a single token worth of activations on the input allows the attacker to determine $\approx 119$ tokens on the output.

**Attacks on activations.** We ran adversarial attacks on model activations right after the embedding layer for a suite of models, a range of attack lengths $a$, target token lengths $t$, and multiple repetitions of each experimental setup (with different random tokens of context $S$ and target $T$ each time), obtaining an empirical probability of attack success $p(a,t)$ for each setting. For multiple repetitions, we also had a standard deviation of $p$ at each set of $(a,t)$ values. To get to lower effective values of $a$ and therefore weaker attacks, we use the multi-attack strategy described in Section 3 and in Fort (2023), designing the same adversarial attack for up to $n = 8$ sequences and targets at once.

Figure 3 shows an example of the results of our experiments on `EleutherAI/pythia-1.4b-v0`, a 1.4B model with activation dimension $d = 2048$. In Figure 3a the success rates of attacks $p$ for different values of attack length $a$ (in tokens whose activations the attacker controls), target length $t$ (in predicted output tokens), and the attack multiplicity

---

[1]`https://huggingface.co/EleutherAI/pythia-70m`
[2]`https://huggingface.co/microsoft/phi-1`
[3]`https://huggingface.co/datasets/roneneldan/TinyStories`

Table 1: A summary of attack multipliers $\kappa$ estimated from experiments for activation adversarial attacks for various language models. $d/\kappa$ is the number of dimensions of an activation needed to control a single output token, while $\chi$ is the attack resistance (defined in Theory 3) which corresponds to the number of typical bits on the input the attacker has to control in order to have a single bit of control on the output. The pythia-* models are from EleutherAI, Phi-* from Microsoft and TinyStories-* from roneneldan.

| Model | Curves | Size | Dimension $d$ | $|V|$ | Attack multiplier $\kappa$ | Per dim multiplier $d/\kappa$ | Attack resistance $\chi$ |
|---|---|---|---|---|---|---|---|
| pythia-70m | Fig 7 | 70M | 512 | 50304 | $24.2 \pm 0.8$ | $21.2 \pm 0.7$ | $21.7 \pm 0.7$ |
| pythia-160m | Fig 8 | 160M | 768 | 50304 | $36.2 \pm 2.0$ | $21.2 \pm 1.2$ | $21.7 \pm 1.2$ |
| pythia-410m-deduped | Fig 9 | 410M | 1024 | 50304 | $70.1 \pm 2.3$ | $14.6 \pm 0.5$ | $15.0 \pm 0.5$ |
| pythia-1.4b-v0 | Fig 3 | 1.4B | 2048 | 50304 | $128.4 \pm 3.7$ | $16.0 \pm 0.5$ | $16.3 \pm 0.5$ |
| pythia-2.8b-v0 | Fig 10 | 2.8B | 2560 | 50304 | $104.9 \pm 2.2$ | $24.4 \pm 0.5$ | $25.0 \pm 0.5$ |
| Phi-1 | Fig 11 | 1.3B | 2048 | 50120 | $42.8 \pm 2.3$ | $47.9 \pm 2.6$ | $49.0 \pm 2.6$ |
| Phi-1.5 | Fig 12 | 1.3B | 2048 | 50120 | $78.4 \pm 5.1$ | $26.1 \pm 1.7$ | $26.8 \pm 1.7$ |
| TinyStories-33M | Fig 13 | 33M | 768 | 50257 | $27.2 \pm 2.2$ | $28.2 \pm 2.3$ | $28.9 \pm 2.3$ |

$n$ (how many attacks at once the same perturbation $P$ has to succeed on simultaneously). The higher the attack length $a$, the more powerful the attack and the longer the target sequence $t$ that can be controlled by it. We fit a sigmoid from Eq. 3 to each curve to estimate the maximum target sequence length, $t_{\max}$, at which the success rate of the attack drops to 50% (an arbitrary value).

In Figure 3b, we plot these $t_{\max}$ maximum target lengths as a function of the attack strength $a$. Since the multi-attack $n$ allows us to effectively go below $a = 1$, we actually plot $a/n$, the effective attack strength. Fitting our scaling law from Eq. 1 justified on geometric and dimensional grounds in Section 2, we estimate the *attack multiplier* for `EleutherAI/pythia-1.4b-v0` to be $\kappa = 119.2 \pm 2.9$, implying that by controlling a single token worth of activations at the beginning of a context, the attacker can determine $\approx 119$ tokens exactly on the output. The results shown in Figure 3 only include experiments with attack lengths $a = 1, 2, 3, 4, 8$ and $n = 1$, and attack length $a = 1$ with a varying $n = 1, 2, 4, 8$. Varying $n$ allows us to go to "sub"-token levels of attack strength.

**Model comparison.** For a number of different models, we show the scaling laws for attack strength $a$ (in tokens whose activations the attacker can control) vs target length $t$ (in tokens) in Figure 6. For each model, we estimate the attack multiplier $\kappa$ (the number of target tokens the attacker can control with a 50% success rate by attacking the activation of a single token on the input), and compute their attack resistance $\chi$, as

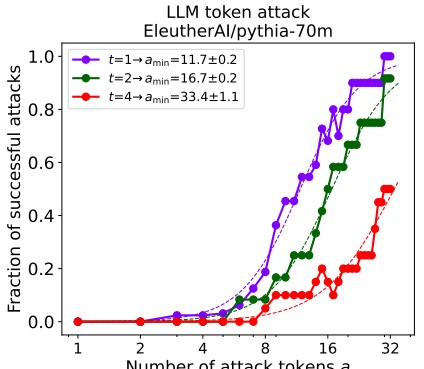

Figure 4: Greedy search over all attack tokens to get specific target completions. In general, $\approx 8$ tokens worth of attack are needed to force a single particular token of a response.

defined in Eq. 3, corresponding to the number of bits the attacker needs to control on the input in order to control a single bit on the output. We summarize these results in Table 1.

An interesting observation is that while the model trainable parameter counts span two orders of magnitude (from 33M to 2.8B), and their activation dimensions range from 512 to 2560, the resulting relative attack multipliers $\kappa/d$, and the attack resistances $\chi$ stay surprisingly constant. This is supporting evidence for our geometric view of the adversarial attack theory presented in Section 2. The conclusion is that for the `EleutherAI/pythia-*` model family, we need between $\approx 15$ and $\approx 25$ bits controlled by the attacker on the model input (the activations of the context) in order to control in detail the outcome of a single bit on the model output (the `argmax` predictions from the model). In an ideal scenario, where a single dimension / bit on the input could influence a single dimension / bit on the output, each activation dimension would be able to control a single token on the output since the model activations are typically 16 bits and to determine a single token we also need $\log_2(V) \approx 16$ bits. However, since we need $\chi > 1$ bits on the input to affect one on the output,

this means that we need $\chi$ activation dimensions to force the model to predict a token exactly as we want. This is still a remarkable strong level of control, albeit weaker than one might naively expect.

**Replacing tokens directly.** To compare the effect of attacking the activation vectors and attacking the input tokens directly by replacing them, we ran experiments on the `EleutherAI/pythia-70m` model. For a randomly chosen context of $s = 40$ tokens and randomly chosen $t$-token target tokens ($t = 1, 2, 4$ in our experiments) we greedily and per-token-exhaustively search over the replacements of the first $a$ tokens of the context in order to make the model predict the desired $t$-token sequence as a continuation. The method is described in Section 3 in detail.

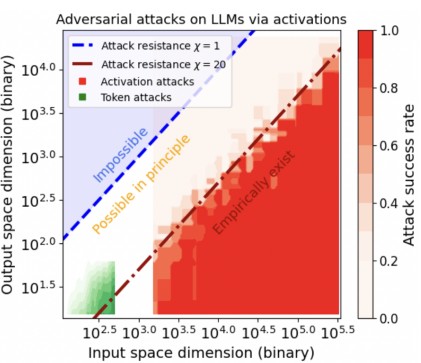

Figure 5: Input-output space dimension comparison for successfully realized adversarial attacks. The regions edge corresponds well to the $d_{\text{in}} \propto d_{\text{out}}$ implication of the dimensional hypothesis.

Unlike for activation attacks, the token replacement attack needs more than one token on the input to influence a single token on the output. The detailed curves showing the attack success rate $p(a, t)$ as a function of the attack length $a$ (the number of tokens the attacker can replace by other tokens) and the target length $t$ are shown in Figure 4, together with Eq. 3 fits to them to extract the $a_{\text{min}}$, the minimum attack length to force the prediction of a $t$-token sequence. For $t = 1$, we get $a_{\text{min}} = 11.7 \pm 0.2$, while for $t = 2$ $a_{\text{min}} = 16.7 \pm 0.2$ and for $t = 4$ $a_{\text{min}} = 33.4 \pm 1.1$. Extracting the attack multiplier $\kappa_{\text{token}} = t/a_{\text{min}}$, we get $0.085 \pm 0.001$, $0.120 \pm 0.002$ and $0.120 \pm 0.004$. Averaging the three estimates while weighting them using the squares of their errors, we get $\kappa_{\text{token}} \approx 0.12$. That means that we need $1/\kappa_{\text{token}} \approx 8$ tokens to control on the input in order to force the prediction of a single token on the output by token replacement (compared to $\approx 0.04$ tokens worth for the activations attack in Table 1).

Controlling tokens instead of activations offers the attacker a greatly diminished dimensionality of the space they can realize the attack in. In Section 2, we discuss the comparison between the dimensionality of the token attack ($\log_2(V)a$ bits of control for an $a$ token attack), compared to the activations attack ($adp$ bits of control for a precision $p = 16$ and $d = 512$ for `EleutherAI/pythia-70m` in particular). Our geometric theory predicts that their attack multipliers should be in the same ratio as the dimensions of the spaces the attacker controls. In this particular case, the theory predicts $\kappa_{\text{token}}/\kappa = \log_2(V)/(dp) \approx 2 \times 10^{-3}$. The actual experimentally estimated values give us $\kappa_{\text{token}}/\kappa \approx 5 \times 10^{-3}$. Given how simple our theory is and how different the activation vs token attacks are, we find the empirical result to match the prediction surprisingly well (it is better than an order of magnitude match).

If we calculated the attack resistance $\chi_{\text{token}}$ for the token attack, we would get directly $\chi_{\text{token}} = 1/\kappa_{\text{token}} = 8.3$. That means that $\approx 8$ bits of control are needed by the attacker on the input to control a single bit of the model predictions. This shows that the token attack is much more efficient than the activations attack ($\chi \approx 22$ for this model), which makes sense given that the model was trained to predict tokens after receiving tokens on the input rather than arbitrary activation vectors not corresponding to anything seen during training.

**Input-output dimension summary** We summarize all our successfully realized attacks in Figure 5. The edge of the region where activation attacks are successful corresponds well to a $d_{\text{in}} \propto d_{\text{out}}$ with an attack resistance (constant of proportionality) of $\chi \approx 20$. Token-based attacks, while shifted to a lower $\chi$, have a similar shape. Notable, all successful attacks lie well below the theoretical boundary of $\chi = 1$ where $d_{\text{in}} = d_{\text{out}}$ and above which attacks should generically not be possible.

## 5 CONCLUSION

Our research presents a detailed empirical investigation of adversarial attacks on language model activations, demonstrating a significant vulnerability that exists within their structure. By targeting a very small amount of language model activations that can be hidden deep within the context window

Table 2: Comparing activation attacks and token attacks on EleutherAI/pythia-70m. The activation attack has a much higher attack multiplier $\kappa = 24.2 \pm 0.8$ compared to the one for token attack of $\kappa_{\text{token}} \approx 0.12$. While a single token's activation can therefore determine $> 20$ output tokens, the attacker needs $> 8$ input tokens to control a single one by token replacement. Our experiments show that they are a factor of $2.6\times$ from each other, which is a close match given the simplicity of our theory and the vastly different nature of the two attack types.

| Model | Attack type | Curves | Size | Dimension $d$ | $\lvert V \rvert$ | Attack multiplier $\kappa$ | Attack resistance $\chi$ |
|---|---|---|---|---|---|---|---|
| pythia-70m | Tokens | Fig 4 | 70M | 512 | 50304 | 0.12 | 8.3 |
| | Activations | Fig 7 | 70M | 512 | 50304 | $24.2 \pm 0.8$ | $21.7 \pm 0.7$ |

and well separated from their intended effect, we have shown that it is possible for an attacker to precisely control up to $\mathcal{O}(100)$ subsequent predicted tokens down to the specific token IDs being sampled (for both randomly sampled and natural language token sequences). The general method is illustrated in Figure 1.

We empirically measure the amount of target tokens $t_{\max}$ an attacker can control by modifying the activations of the first $a$ tokens in the context window, and find a simple scaling law of the form $t_{\max}(a) = \kappa a$, where we call the model-specific constant of proportionality $\kappa$ the *attack multiplier*. We conduct a range of experiments on models from 33M to 2.8B parameters and measure their attack multipliers $\kappa$, summarized in Figure 6 and Table 1.

We connect these to a simple geometric theory that attributes adversarial vulnerability to the mismatch between the dimension of the input space the attacker controls and the output space the attacker would like to influence. This theory predicts a linear dependence between the critical input and output space dimensions for which adversarial attacks stop being possible, which is what we see empirically in Figure 6. By using language models as a controllable test-bed for studying regimes of various input dimensions (controlled by the attack length $a$) and output dimension (controlled by the target sequence length $t$), we were able to explore parameter ranges that were not previously accessible in computer vision experiments where the study of adversarial examples was historically rooted, and sweep a full input dimension – output dimension space of model configurations, as shown in Figure 5.

We find that empirically, the attack multiplier $\kappa$ seems to depend linearly on the model activation dimension $d$ rather than the model parameters, as predicted by our geometrical theory. $\kappa/d$, and as shown in Table 1, is surprisingly constant between $16.0\pm0.5$ and $24.4\pm0.5$ for the EleutherAI/pythia model suite spanning model sizes from 70M to 2.8B parameters.

We compare the activation attacks to the more standard token substitution attacks in which the attacker can replace the first $a$ tokens of the context in order to make the model predict a specific $t$-token sequence. In Figure 4 and Table 2 we summarize our results and show that despite the token attack being much weaker (on our 70M model) with an attack multiplier of $\kappa_{\text{token}} \approx 0.12$ (meaning that we need 8 tokens of input control to make the model predict a single output token) compared to the activation attack $\kappa = 24.2$, the two vastly different attack types have a very similar attack resistance $\chi$ when accounting for the vastly different dimensions of the input space of tokens and activations. The theoretically predicted $\kappa/\kappa_{\text{token}} = 2 \times 10^{-3}$ is surprisingly close to the empirically measured $5 \times 10^{-3}$ despite the arguable simplicity of the theory.

To make sure our findings are robust to a separation between the attack tokens and the target tokens, we experiment with adding up to $\mathcal{O}(1000)$ randomly sampled tokens between the attack and the target in Figure 3c. We find that the attack strength essentially unaffected up to 100 tokens of separation with a logarithmic decline after. However, even a full context of separation gives a high degree of control of the very first token activation over the next-token prediction.

Attacking activations might seem impractical since the majority of language models, especially the commercial ones, allow users to interact with them only via token inputs. However, an increasing attack surface due to multi-model models, where other modalities are added to the activations directly, as well as some retrieval models, where retrieved documents are mixed-in likewise as activations rather than tokens, directly justify the practical relevance of this paper.

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

# A  ATTACK ALGORITHM

---

**Algorithm 1** Computing loss for an activation attack $P$ towards a $t$-token target sequence

---

1: **Given** activation dimension $d$, tokens of context $S$ of length $s = |S|$, attack length $a$ ($a \leq s$), target tokens $T$ of length $t$, and an attack vector $P \in \mathbb{R}^{a \times d}$
2: Compute the activations of the context followed by the target $v = f_{\text{before}}(S + T) \in \mathbb{R}^{(s+t) \times d}$
3: Add the perturbation vector to the first $a$ tokens of $v$ as $v' = v$, $v'[:a] += P$ {This is the attack}
4: **for** $i = 0$ **to** $t - 1$ **do**
5:    Predict logits after the $s + i$ tokens of the context and target $z_i = f_{\text{after}}(v'[:s+i])$
6:    Compute the cross-entropy loss $\ell_i$ between these logits $z_i$ and the target token $T[i]$.
7: **end for**
8: Get the total loss for the $t$-token prediction as $\mathcal{L} = \frac{1}{t} \sum_{i=0}^{t-1} \ell_i$

---

# B  SCALING LAWS FOR VARIOUS MODELS

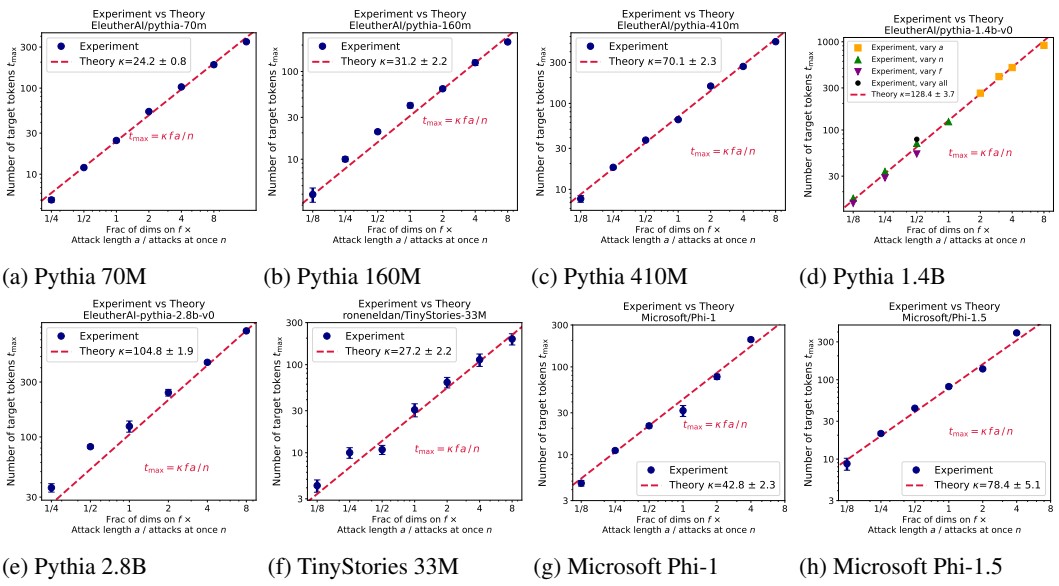

Figure 6: A summary of experiments to determine scaling laws for different models and to read off their attack multiplier $\kappa$. The individual plots show the results of fits to the success rate of attacking a language model by modifying its activations towards the target of generating $t$ tokens as a function of $a$ token activations of an attack. Our theory predicts a linear dependence in each plot, while the slope $\kappa$ (the attack multiplier) is a model-specific constant.

# C  DETAILED ATTACK SUCCESS RATE CURVES FOR DIFFERENT MODELS

# D  EXHAUSTIVE ATTACKS ON INPUT TOKENS

# E  DETAILED EXPERIMENTAL RESULTS

# F  USING ONLY A FRACTION OF DIMENSIONS

Another way of modifying the attack strength is to choose only a fraction $f$ of the activation dimensions the attacker controls. Our geometric theory described in Section 2 suggests that the effective

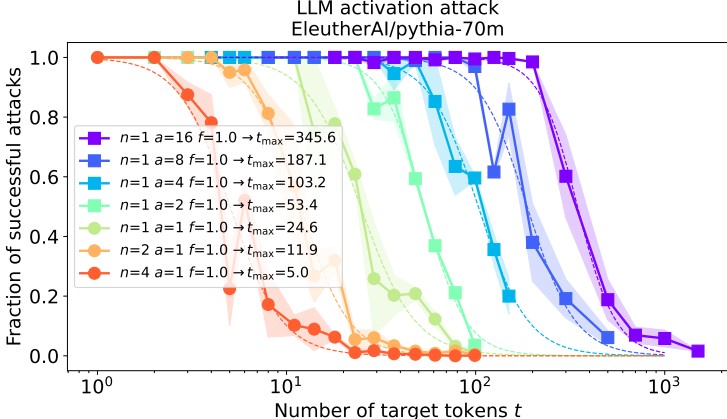

Figure 7: Fraction of successfully realized attack tokens as a function of the number of target tokens $t$ for different numbers of simultaneous attacks $n$ and the length of the attack $a$ for EleutherAI/pythia-70m.

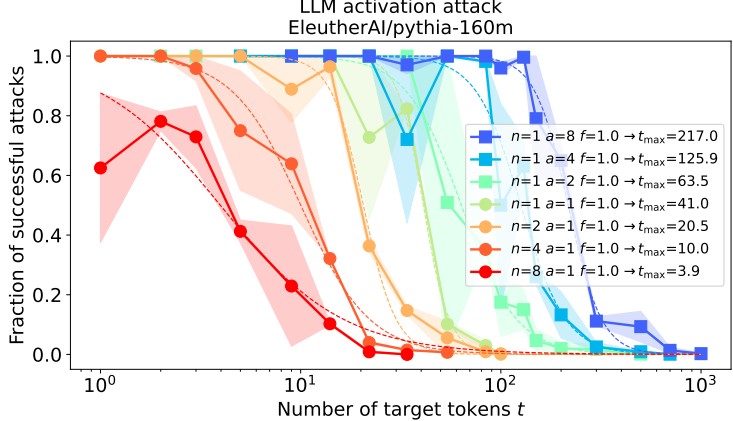

Figure 8: Fraction of successfully realized attack tokens as a function of the number of target tokens $t$ for different numbers of simultaneous attacks $n$ and the length of the attack $a$ for EleutherAI/pythia-160m.

attack strength depends on the product of the attack length $a$, the fraction $f$ and of the attack multiplicity as $1/n$. Therefore we should be able to vary $(f, a, n)$ as we wish and the attack strength should only depend on $fa/n$. In Figure 14 we can see a comparison of experiments at the same attack strength performed at different combinations of $a$, $n$ and $f$. In total 12 experimental setups are shown: 1) $n = 1$ and $f = 1$, while varying $a = 1, 2, 3, 4, 8$, 2) $a = 1$, $f = 1$, and $n = 2, 4, 8$, 3) $n = 1$, $a = 1$, and $f = 1/8, 1/4, 1/2$, and 4) $n = 8$, $a = 8$, and $f = 1/2$. All of these lie on the theoretical predicted scaling law in Eq. 2 as $t_{max} = \kappa fa/n$. The estimated attack multiplier $\kappa = 128.4 \pm 3.7$ is well within $2\sigma$ of the estimate using the varying $a$ and $n$ alone in Figure 3.

## G    CONTEXT SEPARATING THE ATTACK AND THE TARGET TOKENS

In our experiments, we attack the activations of the first $a$ tokens of the context of length $s$ ($a \leq s$) in order to make the model predict an arbitrary $t$-token sequence of tokens as its argmax continuation. In our most standard experiments $a = s$, which means that the attacker controls the activations of the full context which is of exactly the same length as the attack. These are the experiments in Figure 3, summary Figure 6 and the summary Table 1. To see the effect of having

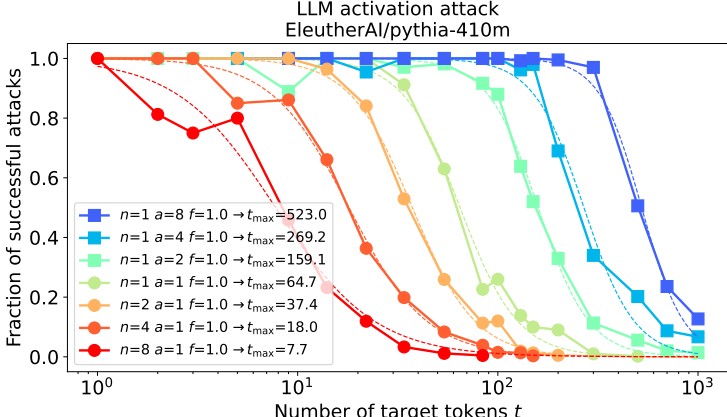

Figure 9: Fraction of successfully realized attack tokens as a function of the number of target tokens $t$ for different numbers of simultaneous attacks $n$ and the length of the attack $a$ for EleutherAI/pythia-410m.

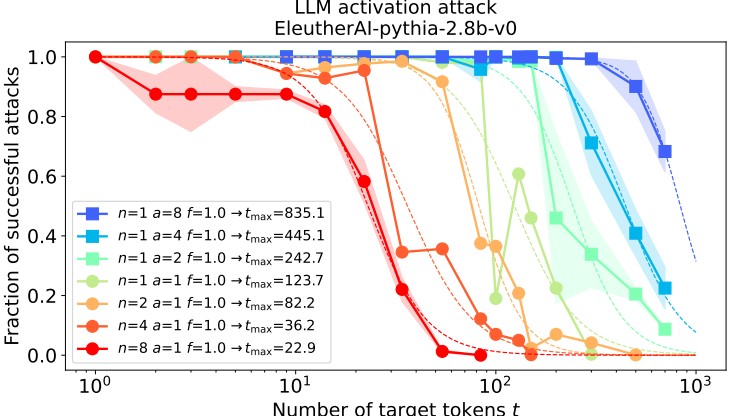

Figure 10: Fraction of successfully realized attack tokens as a function of the number of target tokens $t$ for different numbers of simultaneous attacks $n$ and the length of the attack $a$ for EleutherAI/pythia-2.8b.

context tokens separating the attack from the target tokens, we ran an experiment with a fixed model `EleutherAI/pythia-70m` and read off the attack multiplier $\kappa$ for each context length $s$ in a logarithmically spaced range from 1 to 2000 (almost the full context window size). Estimating each $\kappa(s)$ involved exploring a range of target lengths $a$ and attack multipliers $n$, each in turn being a 300 step optimization of the attack, as described in Section 4. The results are shown in Figure 3c. We see that for up to 100 tokens of separation between the attack and the target tokens, there is no visible drop in the attack multiplier $\kappa$. In other words, the attack is equally effective at forcing token predictions immediately after its own tokens or 100 tokens down the line. After the context length of $\approx 100$ we see a linear drop in $\kappa(s)$ with a log of the context length. At 2000 tokens of random context separating the attack and the target, we still see $\kappa \approx 8$, i.e. a single token's activations on the input controlling 8 tokens on the output.

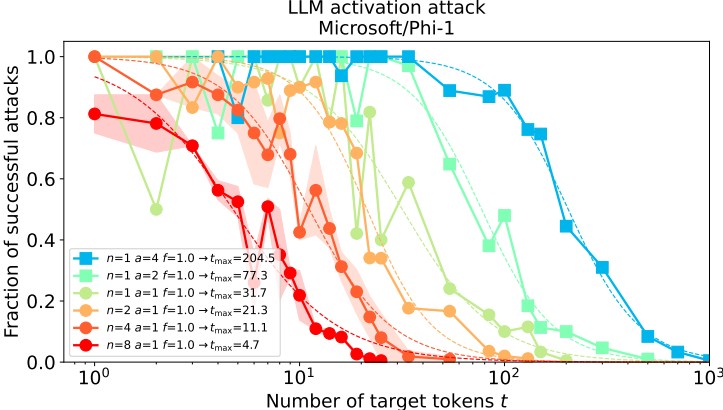

Figure 11: Fraction of successfully realized attack tokens as a function of the number of target tokens $t$ for different numbers of simultaneous attacks $n$ and the length of the attack $a$ for Microsoft/Phi-1.

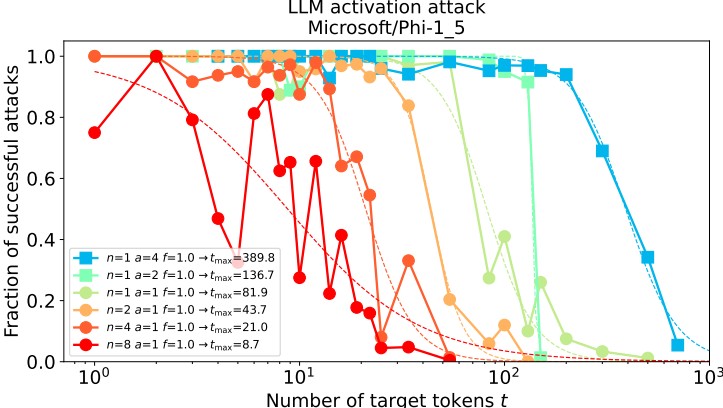

Figure 12: Fraction of successfully realized attack tokens as a function of the number of target tokens $t$ for different numbers of simultaneous attacks $n$ and the length of the attack $a$ for Microsoft/Phi-1.5.

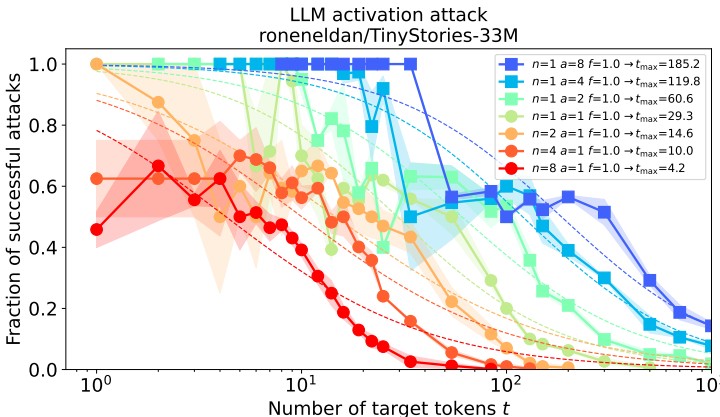

Figure 13: Fraction of successfully realized attack tokens as a function of the number of target tokens $t$ for different numbers of simultaneous attacks $n$ and the length of the attack $a$ for roneneldan/TinyStories-33m.

---

**Algorithm 2** Greedy, exhaustive token attack towards a $t$-token target sequence

---

1: **Given** tokens of context $S$ of length $s = |S|$, attack length $a$ ($a \leq s$), target tokens $T$ of length $t$, and a vocabulary of size $V$
2: The current context sequence starts at $S' = S$
3: **for** $i = 0$ **to** $a - 1$ **do**
4:     {Looping over attack tokens}
5:     **for** $\tau = 0$ **to** $V - 1$ **do**
6:         {Looping over all possible single tokens}
7:         $S'[i] = \tau$ {Trying the new token}
8:         Calculate the language modeling loss $\mathcal{L}_\tau$ for the target continuation $T$ after the context $S'$. If it is lower than the best so far, keep the $\tau_{\text{best}} = \tau$.
9:     **end for**
10:     Update the i$^{\text{th}}$ attack token to the best, greedily, as $S'[i] = \tau_{\text{best}}$ {Greedy sampling token by token. Each token is searched exhaustively.}
11: **end for**
12: If the newly updated context $S'$ produces the continuation $T$ as the result, the attack is successful.

---

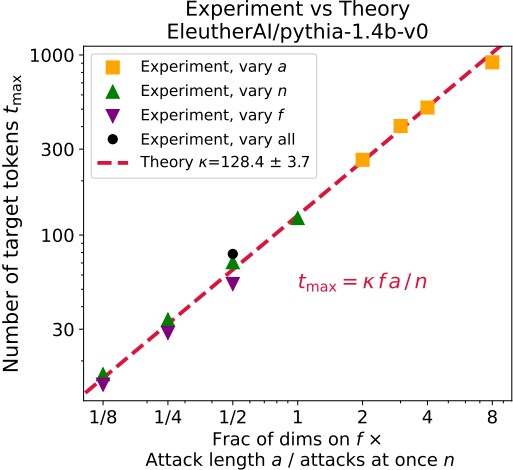

Figure 14: A scaling plot showing successful attacks on Pythia-1.4B for different attack lengths $a$, fraction of dimensions $f$, and attack multiplicities $n$.

