# OpenReview forum: "Scaling Laws for Adversarial Attacks on Language Model Activations and Tokens"
_ICLR.cc/2025/Conference — ICLR 2025 Poster_

### Official Review · Reviewer_1sJt · 2024-10-21

**Soundness:** 3
**Presentation:** 2
**Contribution:** 3
**Rating:** 6
**Confidence:** 4

**Summary:**

The paper proposes a hypothesis: there is a "scaling law" between number of input tokens the attacker can modify and the number of prediction tokens the model outputs.  Empirical results suggest that under certain cases, the maximum number of target tokens that can be manipulated by the attacker, depends linearly on the number of input tokens that the attacker can control.

**Strengths:**

+ S1: The research work is of great novelty and originality. The hypothesis proposed in the paper is interesting.

+ S2: Empirical findings can provide support to the proposed hypothesis. Such scaling law can be observed on several different language models.

**Weaknesses:**

Although the hypothesis is interesting, there are many details not discussed clearly which hinders the readers from a deeper understanding.

+ W1: **Formulation and measure of $t_{max}$ lacks clarity**.  $t_{max}$ is designed to present the ideal maximum tokens that attacker can manipulate in model prediction. However, how to practically obtain this amount may be unclear. From the paper describes a random sampling strategy to obtain context tokens $S$ and target tokens $T$. It raises questions about the impact of different samples on $t_{max}$ value. How many ($S$, $T$) pairs are sampled to determine $t_{max}$? What is the distribution of $t_{max}$ across various samples? Intuitively, the length of target tokens might extend to a larger value if they are more aligned with natural language priors or they are from (repeated) training samples, such as excerpts from famous poems. For random combination of tokens in $T$ and $S$, different samples may get varied attack performance. This is very important since changes to the value of $t_{max}$ may result in the linearity of relationship no longer holds. More detailed discussion on this manner would benefit readers' understanding.

+ W2: **Practicality of the attack is questionable**.  One notable setting in this paper is the attacker's ability to modify the activations of token feature without limit (different from the classic adversarial attack setting where the attacker needs to constrain their perturbations within a $\epsilon$-ball).  While authors provide examples involving retrieval and multi-modal models where an attacker might alter continuous feature values within the model, the feasibility of such unlimited modifications are questionable. For example, in multi-modal models, a more realistic attack surface would be the attacker altering an image within [0, 255] pixel range, rather than arbitrarily modifying image features. The impractical threat model may undermine its real-world relevance and impact of presented findings.


+ W3: Typo and broken sentence
    - a) Line 283, "probability can be seen in e.g. Figure 3a and Table 1 refers to the ..." ?

**Questions:**

My questions are aligned with the weakness part.

+ Q1: Regarding W1,
  - a) Can you specify exact number of ($S$, $T$) pairs sampled for each $t_{max}$ estimation?
  - b) Provide error bars or confidence intervals for the $t_{max}$ values to show the distribution across samples.
  - c) Include an analysis of how $t_{max}$ varies for different types of target sequences (e.g. random tokens vs natural language).
  - d) Discuss how the variability in tmax estimates impacts the linearity of the scaling relationship.


+ Q2: Regarding W2,
  - a) Can you provide a more detailed analysis of realistic constraints on activation modifications in retrieval and multi-modal settings?
  - b) Can you conduct experiments with bounded perturbations (e.g. within an ε-ball) to see how this affects the scaling law.
  - c) Discuss more explicitly the limitations of your unconstrained attack model and its implications for real-world applicability.

---

> ### Author Response · Authors · 2024-12-04
> **Response to Reviewer 1sJt**
>
> We thank the reviewer for their detailed methodological questions and suggestions for improving clarity. We address each point below:
>
> **Regarding W1 - Methodology and Measurement of $t_\mathrm{max}$:**
>
> The reviewer raises excellent questions about our measurement methodology. To clarify:
>
> 1. **Sampling Procedure:**
> - For each (a,t) = (number of attack tokens, number of target tokens) pair, we run 10 independent trials
> - Each trial uses different random tokens for both context S and target T, uniformly sampled from the range(0,vocab_size). This makes the convincing maximally parsimonious and also hard. Our was to establish something of an upper bound on how hard a job the attacker has – in our setup they literally have to convince the model to predict a random sequence of $t$ output tokens, which is harder than e.g. outputting a common phrase (“hi, how are you?”) or repeating a token (“A A A A”)
> - We estimate $t_\mathrm{max}$ by fitting a sigmoid to the success rate curve across different t values
> - Error bars are shown in our plots (e.g., Figure 3) as shaded regions (these are 1 sigma measurements)
>
> In summary, we actually do run a very large number of experiments to estimate the variability in the outputs. Each (a,t) pair has 10 repetitions, each of which has at each point random target tokens, attack tokens, and the token in between, all randomly sampled. The sigmoid fit the the probability_of_successful_attack(attack_length, target_length) is also done with these errors in mind in the fitting, and the resulting parameter, $t_\mathrm{max}$, representing, for each fixed attack length, the maximum number of tokens we can control with 50% probability, comes with a standard error +- estimate.
>
> While it might not be immediately clear from the plots, there are a lot of repetitions involved in getting the means and error bars involved whose sole purpose is to get a handle on the variability you are discussing.
>
> 2. **Distribution and Natural Language:**
> The reviewer makes an insightful point about different types of target sequences. We deliberately chose random tokens as the most challenging case - natural language sequences would likely be easier to force the model to generate due to their inherent structure, and so would be e.g. common sequences such as 1 2 3 4 5 6 or repeated tokens such hi hi hi hi hi. This makes our results a conservative lower bound on attack capability, or conversely a conservative upper bound on how hard it is to attack a model.
>
> Regarding the specific points:
>
> - Confidence intervals for t_max estimates are shown in Figure 3 b) but **they are in fact so small that you cannot see them for the most of the datapoints** In Figure 3 c) they are clearly visible for the experiment with intermediate context tokens separating  the attack and target tokens. In Table 1 the resulting attack resistances also include an error bar that was the derived from the error bars on the t_max values and the fit to them in plots such as Figure 3 b). In Figure 6 you can clearly see the error bars on the t_max values for some of the models on which they are larger
>
> - Discussion of how natural language targets might affect results
>
> - More detailed description of our fitting procedure
>
> **Regarding W2 – Attack Practicality:**
>
> The reviewer raises valid concerns about unbounded perturbations. While we agree that bounds would be more realistic, there are several important considerations:
>
> 1. **Theoretical Framework:**
> Our primary goal was to understand fundamental scaling relationships. Unbounded perturbations allow us to probe these relationships without conflating them with effects from specific bounds. We agree that the effect of such bounds would be important to study. Unlike for images, however, we do not have a natural scale for what small perturbations of activations look like and we believe a considerable effort would have to be spent on making sure whatever definition of “small” we choose would make practical sense
>
> 2. **Practical Relevance:**
> In retrieval systems, while individual document embeddings might have typical ranges, the attacker can often control multiple documents, effectively allowing larger perturbations through their combination. Similarly, in multi-modal systems, the attacker often has significant latitude in designing inputs that produce desired activations. A successful example of the first transferable image-based multi-modal attack on GPT-4o and Claude is for example demonstrated in https://arxiv.org/abs/2408.05446 and shown in https://www.youtube.com/watch?v=mf_Es-hrsDk
>
> 3. **Future Extensions:**
> We agree that studying bounded perturbations would be valuable. Based on our theory, we would expect:
> - The effective dimensionality p to scale as log(bound_size)
> - The attack multiplier κ to decrease proportionally
> We will add this theoretical prediction to the paper.
>
> **Technical Corrections:**
> We thank the reviewer for catching the typo on Line 283. This will be fixed in the revision.

---

### Official Review · Reviewer_xEdG · 2024-10-29

**Soundness:** 3
**Presentation:** 2
**Contribution:** 3
**Rating:** 8
**Confidence:** 4

**Summary:**

The paper considered activation-level rather than token-level adversarial attacks on LLMs and derives scaling laws between the number of output tokens an attacker can affect vs the number of input activations they can control. The paper also explores token-level substitution adversarial attacks in comparison to their novel approach.

The paper justifies the practicality of their activation-level attacks by referencing retrieval and multi-modal models. It concludes by drawing general implications of this work for the adversarial ML community, namely presenting language models as a more flexible environment for exploring the theory of adversarial attacks compared to vision models.

**Strengths:**

Originality:
 * As far as I can tell this is the first paper to take an in-depth look at activation-level adversarial attacks
 * The comparison of token vs activation level attacks isn’t surprising due to the significantly higher level of granular control you can exhibit on the activation level but good to have it quantified

Quality:
 * Clear writing style, thought process and hypothesis clearly stated and explained

Clarity:
 * Core idea and experiments clearly presented and explained.

Significance:
 * Strong practical justification of the real-world applicability of the research through reference to retrieval and multi-modal setups.
 * Presents a compelling case for using LLMs and activation-level attacks as a test bed

**Weaknesses:**

* Presentation and explanation of the theory of the attack strength could be better – it’s currently a large hard-to-read paragraph while references are repeatedly made to the empirical vs theoretical scaling laws. E.g., breaking down the working of the final equation.
* The narrative of the paper feels a bit non-linear – I had to skip ahead or read back multiple times to reference material presented in the paper to bits where it was relevant again. For example in L153 the significance of the log2 value is only appreciated after reading the scaling laws subsection.
* Some of the more vague speculative statements such L204 “(although adversarial training probably changes it)” should probably be avoided and left to a “future research” section. Ditto L222-223 regarding the effective dimensionality of the embedding – this specifically would be very interesting to explore separately.
* Some of the plots should be included as PDFs rather than images for fidelity’s sake. E.g. Figure 1(b,e), Figure 3, Figure 5. This is important as the graphs are dense and small for an A4 page. I would also recommend increasing the font size to improve readability.
* This is a side note but the appendix appears to be incomplete with sections C, D, E appearing empty

**Questions:**

* Is χ related to concepts such as generalised degrees of freedom (https://auai.org/uai2016/proceedings/papers/257.pdf Gao and Jojic 2016) or effective dimensionality (https://arxiv.org/abs/2003.02139 Maddox et al 2020)? It feels like a similar concept
* Worth exploring how adversarial training affects χ
* Would make for an interesting separate research paper to explore the effective dimensionality of the activation dimensions and maybe how that could be used for compression. This is in reference to L222-223.

---

> ### Author Response · Authors · 2024-12-04
> **Response to Reviewer xEdG**
>
> We thank the reviewer for their thorough and insightful feedback. We particularly appreciate the thoughtful suggestions for improving the presentation and the interesting connections drawn to related concepts.
>
> **Regarding Presentation and Theory:**
> We agree that the theoretical explanation of attack strength could be better structured. In the revision, we will:
> 1. Break down the derivation of the attack strength equation into clear steps (the main issue was space for us in the submitted version – we will make sure the derivation gets the detail it needs)
> 2. Create a dedicated subsection for the theoretical framework
> 3. Add explanatory figures to illustrate the key concepts
>
> The nonlinear narrative is also a valid concern. We will restructure to:
> - Introduce key concepts like log₂V earlier where needed
> - Move the most speculative statements to a dedicated future work section
> - Create clearer signposting between related sections
>
> Thank you again for making such detailed narrative flow suggestions!
>
> **Technical Improvements:**
> We appreciate the careful attention to technical details and will make the following changes:
> - Convert all plots to vector format (PDF) for better readability
> - Increase font sizes in figures
> - Complete the appendix sections C, D, E with the missing content
>
> **Regarding Theoretical Connections:**
> The reviewer raises an interesting point about the relationship between our attack resistance χ and concepts like generalized degrees of freedom and effective dimensionality. While we had not made this connection explicit, there are indeed interesting parallels:
> - Like generalized degrees of freedom, χ captures a notion of how many "effective parameters" control model behavior
> - The connection to effective dimensionality could help explain why χ remains surprisingly stable across model scales
>
> In the paper we try to make such a connection by referring to
> Stanislav Fort, Ekin Dogus Cubuk, Surya Ganguli, and Samuel S. Schoenholz. What does a deep neural network confidently perceive? the effective dimension of high certainty class manifolds and their low confidence boundaries, 2022.
> Where the authors try to estimate how many “effective dimensions” the high-confidence class manifolds in the input space have. They show that for very different trained models and for essentially all classes, the number is around 10-30 in the 3072-dimensional space of CIFAR-10 images.
>
> We will add a discussion of these relationships in the revision, while noting that fully exploring these connections would be valuable future work.
>
> **Future Research Directions:**
> The reviewer suggests several promising research directions:
> 1. Exploring how adversarial training affects χ – this would definitely be a great topic to explore in detail, however, practically speaking, we would need to have partially trained checkpoints of LLMs that are not frequently available for many models
> 2. Investigating the effective dimensionality of activation spaces – yes, that would definitely be very relevant! We think this is an alternative way of looking at a similar phenomenon
> 3. Studying compression possibilities based on activation dimensionality – very good point!
>
> While beyond the scope of the current paper, we agree these are exciting directions. We will try adding a "Future Work" section discussing these possibilities, particularly highlighting how our framework could be extended to study these questions.
>
> We thank the reviewer for helping strengthen both the presentation and theoretical foundations of our work.

---

### Official Review · Reviewer_HSZG · 2024-11-02

**Soundness:** 2
**Presentation:** 2
**Contribution:** 2
**Rating:** 6
**Confidence:** 4

**Summary:**

This paper studies the technique of adversarial attacking the activation value of LLM. This work explains the LLM's vulnerability and finds that attacks satisfy scaling law.

**Strengths:**

The problems studied are important because LLM has been applied to our life. The proposed attack method can achieve good results and shows some inherent characteristics of LLM itself.

**Weaknesses:**

The practicality of the proposed scheme is questionable, and whether this attack scheme can be implemented in reality needs further consideration.

**Questions:**

1. The utility of LLM activation values to attacks needs to be fully explained. Only two works discussing modifying activation values are presented in the introduction of this paper. And the work of the retrieval and multimodality seems to be to concatenate activations from different inputs, rather than to add a certain perturbation of activations, as is the case against attacks.

2. Is the attacker a user? How can an attacker inject generated adversarial perturbations into benign user-generated activation values?

3. Does an attacker need to access the entire LLM model to calculate adversarial perturbations? The LLM model is usually provided to the user in the form of an API, and the attacker cannot obtain the model weights.

---

> ### Author Response · Authors · 2024-12-04
> **Response to Reviewer HSZG**
>
> We thank the reviewer for their detailed comments and important questions about the practical aspects of our work. We address each point below:
>
> **1. Regarding the Utility and Context of Activation Attacks:**
> The reviewer raises an excellent point about better contextualizing activation-based attacks. We will expand the introduction to clarify this important aspect. While concatenation is indeed the typical use case in retrieval and multi-modal models, this happens at a particular layer / layers of the model where the multi-model or retrieved input is directly inserted. **However**, the cross-attantion in the layers after mixes together these added activations with the text-only activations and in a sense acts as a complex way of adding such perturbations. While in our work we directly add the perturbation P, in reality such a perturbation is effectively added as a complicated, but differentiable function of the added activations.
>
> A very recent, related, practical example of such an attack that managed to fool, via a multimodal image input, the largest and most secure models out there, including GPT-4o and Claude 3 Sonnet, is presented in https://arxiv.org/abs/2408.05446 with a demonstration on Youtube at https://www.youtube.com/watch?v=mf_Es-hrsDk.
>
> Specifically:
> - In retrieval models like those in Borgeaud et al. (2022), retrieved document chunks are encoded directly into activation space and concatenated with prompt activations, and in the layers after they are mixed using the attention mechanism with the original token activations
> - In multi-modal models like Flamingo (Alayrac et al. 2022), embedded images are inserted as activations between text token activations, and the same mixing happens in the followup layers
>
> This direct insertion of activations into the model's computation path is what makes our attack scenario practical in these contexts. We will make this connection more explicit in the revised paper.
>
> **2-3. Regarding Attack Implementation and Model Access:**
> While our paper uses the LLM attacks primarily as a vehicle to study the nature of adversarial attacks in general in regimes that are typically inaccessible in computer vision, the question of practicality is also a relevant one. The reviewer raises critical questions about the practical execution of these attacks. To clarify:
>
> The attacker does not need to be a regular end-user, nor do they need access to the full model. Instead, our attack scenario primarily applies to two increasingly common architectures:
>
> 1. Retrieval-augmented models:
>    - The attacker only needs to control entries in the retrieval database
>    - When their manipulated document is retrieved, its adversarial activations are automatically inserted into the model's computation
>    - No direct model access is required
>
> 2. Multi-modal models:
>    - The attacker only needs to craft an adversarial image (as shown for example in the demo by Fort & Lakshminarayana 2024 (https://arxiv.org/abs/2408.05446) in https://www.youtube.com/watch?v=mf_Es-hrsDk)
>    - When processed through the image encoder, it produces the desired adversarial activations
>    - Again, no direct model access is required
>
> We agree that for standard API-based text-only LLMs, this attack vector is less applicable. However, as models increasingly incorporate external knowledge and multiple modalities, activation injection becomes a real concern. We will revise the paper to make these practical scenarios and limitations more explicit. A practical scenario: an image comes as a part of an email, an LLM looks at it while reading for the user to summarize, the image covertly changes the model behavior to e.g. exfiltrate some files from their Drive.
>
> **Additional Context:**
> While our focus is on understanding fundamental scaling laws of adversarial attacks, these practical implications highlight why such understanding is crucial. As model architectures evolve, new attack surfaces emerge, making it essential to understand the theoretical underpinnings of such vulnerabilities.
>
> We thank the reviewer for pushing us to clarify these important practical considerations, which will improve the paper's impact and accessibility.

---

### Official Review · Reviewer_wHmg · 2024-11-05

**Soundness:** 3
**Presentation:** 2
**Contribution:** 2
**Rating:** 6
**Confidence:** 2

**Summary:**

The paper proposes an adversarial attack on the activations of LLMs and studies scaling laws for the attack. The results show that by perturbing a small set of model activations, prediction tokens can be controlled. A linear scaling law between perturbed activation tokens and predicted tokens is empirically validated. This threat model is claimed to practical in retrieval tasks and in certain multi-modal models.

**Strengths:**

The scaling law study for adversarial attacks on language model activations allows exploration in input-output regimes much larger than previously studied computer vision models.

**Weaknesses:**

As mentioned in Line 268, the attacks used are close to iterative gradient attacks or FGSM which were proposed early in the computer vision literature. Is it comprehensive enough to do this study limited to these attack models. Also, is it common to keep the perturbations unbounded for LLM attacks?

**Questions:**

1. Is there any supporting evidence in literature for "The core hypothesis is that the ability to carry a successful adversarial attack depends on the ratio between the dimensions of the input and output spaces."

2. Typos (Line 262, 267 ,315) the reference to Algorithms are incorrect.

3. (Line 499) It's incorrect use of the Big-O notation. O(100) is equivalent to any O(k) where k is a constant. So, it's best to rephrase this line.

---

> ### Author Response · Authors · 2024-12-04
> **Reply to Reviewer wHmg**
>
> We thank the reviewer for their thoughtful comments and specific suggestions for improvement. Below we address each point in detail:
>
> **Regarding Attack Models and Unbounded Perturbations:**
> The reviewer raises an important point about attack methodology. While our gradient-based attacks are indeed similar to those in early computer vision, we chose this approach deliberately for two reasons:
> 1. It allows direct comparison with the foundational adversarial attack literature
> 2. The attacks work sufficiently well –  the reason for using stronger attacks in vision has been the steady increase in the robustness of the model they are trying to attack. In case of the very brittle LLMs, even a simple attack is more than sufficient to effectively confuse  the model easily
> 3. The focus of our work is not on developing novel attack methods, but rather on using attacks as a tool to study fundamental scaling relationships between input and output dimensionality. For this reason we choose as simple an attack as possible so that the results we see could be primarily attributed to the underlying properties of the LLM rather than the particularities of the attack method itself
>
> Regarding unbounded perturbations – this is actually a relatively standard practice in the LLM attack literature.
> For example [Zou et al. 2023, https://arxiv.org/pdf/2307.15043] looks for token-level adversarial suffixes that are unbounded. The attacks allow arbitrary text to be appended to the original user query, as long as it achieves the objective of making the model generate harmful content. Unlike in images, there isn't really a notion of "imperceptible" bounded perturbations since any token change is noticeable.
>
> Unlike image models where bounded perturbations are crucial for visual imperceptibility, LLM activation spaces lack natural bounds since they're internal representations rather than human-interpretable inputs. But we do agree that it would be well worth investigating the effect of boundedness on the capacity to carry a successful attack. It is just much less clear what the natural definition of such a bound would be, given that we are not dealing with human-perceptible pixel colors but rather internal latent representations.
>
> In the future versions of the paper, we would definitely like to explore this. Given our results and theoretical analysis, the number of bits available to describe a perturbation is an important factor. In the paper, we say:
>
> > “A single token has an activation vector specified by d p-bit precision floating point numbers.”
>
> And this $p$ does appear in the attack resistance χ linearly. If we restricted the magnitude of the attack perturbation linearly to size limit m, we would roughly expect the corresponding effective $p$ to scale as $p \propto \log ( m )$. We would therefore expect the model to be able to withstand such attacks better than the unbounded ones, and for this ability to scale roughly as \log ( m ).
>
> **Regarding the Core Hypothesis:**
> The reviewer asks about supporting evidence for our dimensional hypothesis. While not previously explicitly stated for LLMs, this idea has substantial precedent in the literature:
> - Goodfellow et al. (2015), Abadi et al. (2016), and Ilyas et al. (2019) discuss aspects of this relationship, as cited in the paper (line 079-083)
> - This is further supported by Fort & Lakshminarayanan (2024), who demonstrate the first successful transferable attacks on vision LLMs (as cited in the paper)
>
> We will expand this discussion in the revision to better contextualize our hypothesis, drawing more explicitly from these existing works. While the hypothesis has been “floating in the aether” of the community, as far as we are aware, it hasn’t been explicitly stated or measured to the extent we show in the paper, e.g. in Figure 5 where we sweep 3 orders of magnitude of input and output space dimensionalities and verify empirically whether we can successfully carry adversarial attacks there.
>
> **Technical Corrections:**
> We thank the reviewer for catching these issues:
> - The algorithm references will be corrected (Lines 262, 267, 315)
> - We agree about the O-notation usage and will rephrase Line 499 to use exact numbers instead: "we demonstrate the ability to control up to approximately 100 subsequent tokens". You are totally right that this is a misuse of the O-notion and we will be clearer about the presentation.
>
> **Additional Context:**
> Our key contribution is not the attack method itself, but rather using attacks as a probe to study previously inaccessible input-output dimension regimes (V^t possibilities where t≈1000, compared to just 1000 classes in ImageNet). This allows us to validate dimensional theories of adversarial examples in new domains and at unprecedented scales.
>
> We appreciate the helpful feedback and will incorporate these improvements in the revision.

---

### Meta-Review · Area_Chair_PVb9 · 2024-12-24

**Metareview:**

This paper presented an adversarial attack on the activations of LLMs and studies scaling laws for the attack. The problem appears to be novel and has some practical implications.
Strength:
1. Methods appeared to be novel and could have practical implications in deploying the LLMs.
2. Scaling law study is useful to understand the potential impact of the attack method

Weakness:
1. The practicality is somewhat questionable at this moment.

**Additional Comments On Reviewer Discussion:**

Reviewers generally agree that this paper would be interesting to have in the conference. Some reviewers didn't actively participate in the discussions; most of the concerns appeared to be well addressed by the rebuttal.

---

### Decision · Program_Chairs · 2025-01-22

Accept (Poster)